# Minocycline inhibits rosacea-like inflammation through the TLR4-mediated NF-κB signaling pathway in vivo and in vitro

Piyan Hua☯, Ying Tu☯, Zhenghui Yang, Yunting He, Li He, Qiuyan Yao, Hua Gu🆔*

Department of Dermatology, First Affiliated Hospital of Kunming Medical University, Kunming, Yunnan Province, China

☯ These authors contributed equally to this work.
* guhua1978@sina.com

## Abstract

### Background

Rosacea is a chronic inflammatory skin disease characterized by multiple intricate pathogenic factors. Previous studies have substantiated the anti-inflammatory properties of minocycline and its potential therapeutic efficacy in treating rosacea. However, further elucidation of the underlying mechanism is warranted.

### Methods

HaCaT cells and BALB/c mice were treated with LL37. Moreover, the effect of minocycline on rosacea was explored through the addition of an NF-κB inhibitor (PDTC) or overexpression of Toll-like receptor 4 (TLR4). The expression of related markers was detected by western blotting, immunofluorescence, ELISA, flow cytometry, etc.

### Results

Minocycline suppressed dermal infiltration of inflammatory cells in rosacea-like mice and reduced the expression of inflammatory cytokines in rosacea-like mice and cells. Moreover, minocycline downregulated the expression of TLR4 and p-NF-κB thereby inhibiting ROS production. However, overexpression of TLR4 or the addition of PDTC counteracted the effects of minocycline by promoting cellular inflammation and ROS production. Mechanistically, minocycline hinders TLR4/TNF-α activation induced by LL37 in skin and cells to suppress the expression of inflammatory cytokines.

### Conclusion

Minocycline alleviates inflammation progression in rosacea by downregulating TLR4 and inhibiting the activation of the NF-κB pathway, providing a scientific basis for subsequent clinical treatment.

**Data availability statement:** All data files are available from the Figshare repository (DOI：https://doi.org/10.6084/m9.figshare.28464782).

**Funding:** This work was supported by the Yunnan Revitalization Talent Support Program (Recipient: Hua Gu).

**Competing interests:** The authors have declared that no competing interests exist.

## 1 Introduction

Rosacea is a recrudescent chronic inflammatory skin condition that is characterized by recurrent episodes of erythema, telangiectasia, and papulopustular lesions involving predominantly on the convexities of the central face [1]. However, the etiology and pathogenesis of rosacea have not been fully elucidated. Currently, it is believed that genetics, immune and neurovascular dysregulation, microbial infection, skin barrier dysfunction, and other factors play important roles in the occurrence and exacerbation of rosacea [2]. Due to its tendency to recur easily along with poor compliance and limited treatment efficacy, the quality of life for patients with rosacea is often significantly impacted [3]. Though there is no cure for rosacea, its manifestations may be reduced or controlled with a series of topical, and oral therapies, light devices, appropriate skincare, and lifestyle management [4].

Minocycline, a second-generation derivative of the tetracycline family, possesses antibacterial, anti-inflammatory, immunoregulatory, neuroprotective, and anti-angiogenic properties. It is commonly utilized for the treatment of respiratory infections, genital tract infections, and skin disorders such as acne and rosacea [5–10]. *Eady EA et al.* [11] have confirmed that minocycline effectively improves acne through its regulation of inflammatory cytokines. β-Hydroxybutyrate and minocycline demonstrate synergistic efficacy in attenuating heat stress-induced neuroinflammation through targeted modulation of the Toll-like receptor 4 (TLR4)/p38 mitogen-activated protein kinase (MAPK) and Nuclear Factor-κ B (NF-κB) signaling cascades [12]. A meta-analysis has shown that 100mg minocycline in the treatment of rosacea was more effective than 40mg doxycycline, 40mg minocycline, topical ivermectin, and 0.75% metronidazole [13]. However, the specific mechanism of minocycline treatment of rosacea has not been clarified.

Toll-like receptor 4 (TLR4) plays a pivotal role in the signaling pathways associated with chronic inflammation [14]. TLR4, as an intact transmembrane protein, initiates downstream signaling cascades via kinases to activate transcription factors like Activator Protein-1 (AP-1) and NF-κB, thereby exerting its specific function in inflammation through the NF-κB signaling pathway [15]. The NF-κB signaling pathway has two distinct activation pathways, with the canonical pathway inducing the expression of genes that regulate immune and inflammatory responses [16]. In epithelial cells, NF-κB plays a significant role in maintaining skin immune homeostasis. A study by *Pasparakis M et al.* [17] confirmed that IkappaBkinase (IKK)/NF-κB signaling in epidermal keratinocytes is crucial for regulating skin immune homeostasis. Multiple studies have indicated a close association between the TLR4/NF-κB signaling pathway and inflammatory response [18,19]. However, the precise molecular mechanism underlying how the TLR4/NF-κB signaling pathway mediates skin inflammation in rosacea remains incompletely understood. Here, the potential therapeutic role and the possible mechanism of minocycline on rosacea in vivo and in vitro have been investigated in this study.

## 2 Methods

### 2.1 Cell culture

HaCaT cells (immortalized human keratinocyte cell line) were purchased from the American Type Culture Collection (ATCC) and incubated in DMEM (Gibco, USA) supplemented with 10% fetal calf serum at 37 °C and 5% $CO_2$ and 95% humidity in the incubator.

### 2.2 Cell transfection

TLR4 overexpression plasmid and its negative control were purchased from GeneChem (Shanghai, China). Lipofectamine 2000 (Invitrogen) was used to transfect cells. The overexpression efficiency of the constructs was confirmed by western blotting 48 h after transfection, after which the oe-TLR4 cells were used in subsequent experiments.

### 2.3 Cell grouping

HaCaT cells in the Negative Control (NC) group were cultured normally without any intervention. One group of HaCaT cells was stimulated with Leucine-Leucine-37 (LL37) at 8 µM for 1 h. The cells in the LL37 + Minocin group were treated with 10 µM minocycline for 2 h and then stimulated with 8 µM LL37 for another 1 h. The oe-TLR4 cells in the LL37 + Minocin+oe-TLR4 group were treated the same as the LL37 + Minocin group. For the LL37 + Minocin+oe-TLR4 + PDTC group, NF-κB inhibitor (PDTC, 60 µmol/L) was added in the oe-TLR4 cells and then treated as the LL37 + Minocin group.

### 2.4 Animal

Seven-week-old BALB/c mice were purchased from Shanghai SLAC Experimental Animal Co. Ltd. (Shanghai). All animal experiments were carried out under specific pathogen-free conditions. In the process of feeding mice, keep the feeding environment clean, and quiet, temperature (22 ± 2) °C, humidity (50 ± 5) % constant, and provide adequate food and clean drinking water, to reduce the stress and pain caused by external environmental factors on mice. All studies and experimental procedures were approved by the Animal Ethics Committee of The First Affiliated Hospital of Kunming Medical University(kmmu20231144). The mice were divided into five groups (the NC group, the LL37 group, the LL37 + Minocin group, the LL37 + Minocin+oe-TLR4 group, and the LL37 + Minocin+oe-TLR4 + PDTC group) and treated differently. The mice were shaved 24 h before treatment and then injected subcutaneously with 40 µL of LL37 peptide twice a day for 3 days. The mice in the LL37 + Minocin group were administered an intraperitoneal injection of minocycline at a dose of 50 mg/kg/day for 3 days. For the LL37 + Minocin+oe-TLR4 group, Minocin-treated mice were injected with 100 µg of oe-TLR4 plasmid via the tail vein. For the LL37 + Minocin+oe-TLR4 + PDTC (5108–96–3, AbMole) group, the mice were intraperitoneally injected with PDTC at a dose of 50 mg/kg. The experiment was performed 72 h after the initial injection of LL37. In order to alleviate the pain of mice, mice were anesthetized by intraperitoneal injection of 1% sodium pentobarbital (40mg/kg) before the experimental operation. After the experiment, the mice were euthanized by cervical dislocation. Skin inflammation was evaluated by the severity of erythema and edema. Skin tissues were collected for protein extraction, hematoxylin-eosin staining, and immunofluorescence analysis.

### 2.5 Western blot

Cells and skin tissues were collected, total proteins were extracted using RIPA buffer containing 1% protease inhibitors, and the total protein concentration was determined by the BCA method. The proteins were heated in a boiling water bath for 5 min to fully denature the proteins, after which the target bands were separated via 10% SDS–polyacrylamide gel electrophoresis. After electrophoresis, the proteins were transferred to PVDF membranes using a Bio-Rad standard wet transfer apparatus. After the membrane was completely transferred, the membranes were blocked with 5% skim milk for 1 h at room temperature by shaking slowly on a shaker. After blocking, the primary

antibodies anti-TLR4 (1:1000, ab217274, Abcam), anti-NF-κB (1:1000, ab207297, Abcam), anti-p-NF-κB (1:1000, ab76302, Abcam) and anti-β-actin (1:1000, ab8226, Abcam) were diluted at 4 °C in a shaker with gentle shaking overnight. The next day, the HRP-labeled secondary antibody (1:5000, ab205718, Abcam) was added at an appropriate concentration, and the samples were incubated at room temperature with gentle shaking for 1 hour. The protein bands were visualized by enhanced chemiluminescence (ECL) solution and analyzed by ImageJ. The experiment was repeated three times.

## 2.6 Immunofluorescence staining

The expression of tumor necrosis factor-α (TNF-α) in cells and skin tissues was detected via immunofluorescence staining. Cells and frozen skin tissue sections were fixed with 4% paraformaldehyde (Thermo Fisher Scientific, Shanghai, China) for 20 min, blocked with PBS containing 5% bovine serum albumin for 1 h, and incubated with primary antibodies anti-TNF-α (1:250, ab96879, Abcam) overnight at 4 ° C, and then incubated with FITC-labeled secondary antibodies at room temperature for 1 h. The nucleus was stained with DAPI for 30 min, and the results were imaged by a fluorescence microscope.

## 2.7 Reactive oxygen species (ROS) detection

The expression of ROS in cells was detected by fluorescence staining according to the instructions of the ROS detection kit (Thermo Fisher Scientific, Shanghai, China). The level of tissue ROS was detected by flow cytometry and analyzed by FlowJo software.

## 2.8 Enzyme-linked immunosorbent assay (ELISA)

The levels of TNF-α, Interleukin-6(IL-6), Interleukin-1α(IL-1α), and Interleukin-1β(IL-1β) in cells were quantified using an ELISA kit (Thermo Fisher Scientific, Shanghai, China) following the manufacturer's instructions. The absorbance of both standards and samples was measured using a microplate reader, and the expression levels were calculated according to the manufacturer's instructions.

## 2. 9 HE staining

The mouse skin tissues were fixed in 4% paraformaldehyde, then embedded in paraffin, dehydrated, and sectioned into slices. The tissue sections were stained with hematoxylin or Eosin solution for observation and analysis.

## 2.10 Statistical analysis

All the experimental data in this paper are expressed as the mean ± standard deviation (mean ± SD). GraphPad Prism 7 was utilized for data analysis and visualization. A t-test was employed for comparisons between two groups, while one-way ANOVA was used for comparisons among multiple groups. Pairwise comparisons between groups were conducted using two-way ANOVA. Statistical significance was defined as $P < 0.05$.

## 3 Results

### 3.1 Minocycline suppressed the expression of NF-κB signaling pathway-associated proteins and downstream inflammatory signaling factors

To further investigate the role of minocycline on rosacea, Western blot analysis was employed to assess the expression levels of TLR4 and p-NF-κB/ NF-κB. Compared to the NC group, upregulation of TLR4 and p-NF-κB/ NF-κB expression was observed in both LL37-treated HaCaT cells and skin tissue from rosacea-like mice, which was partially attenuated following minocycline treatment (Figs 1A and 2B). The levels of inflammatory factors such as TNF-α, IL-6, IL-1α, and IL-1β

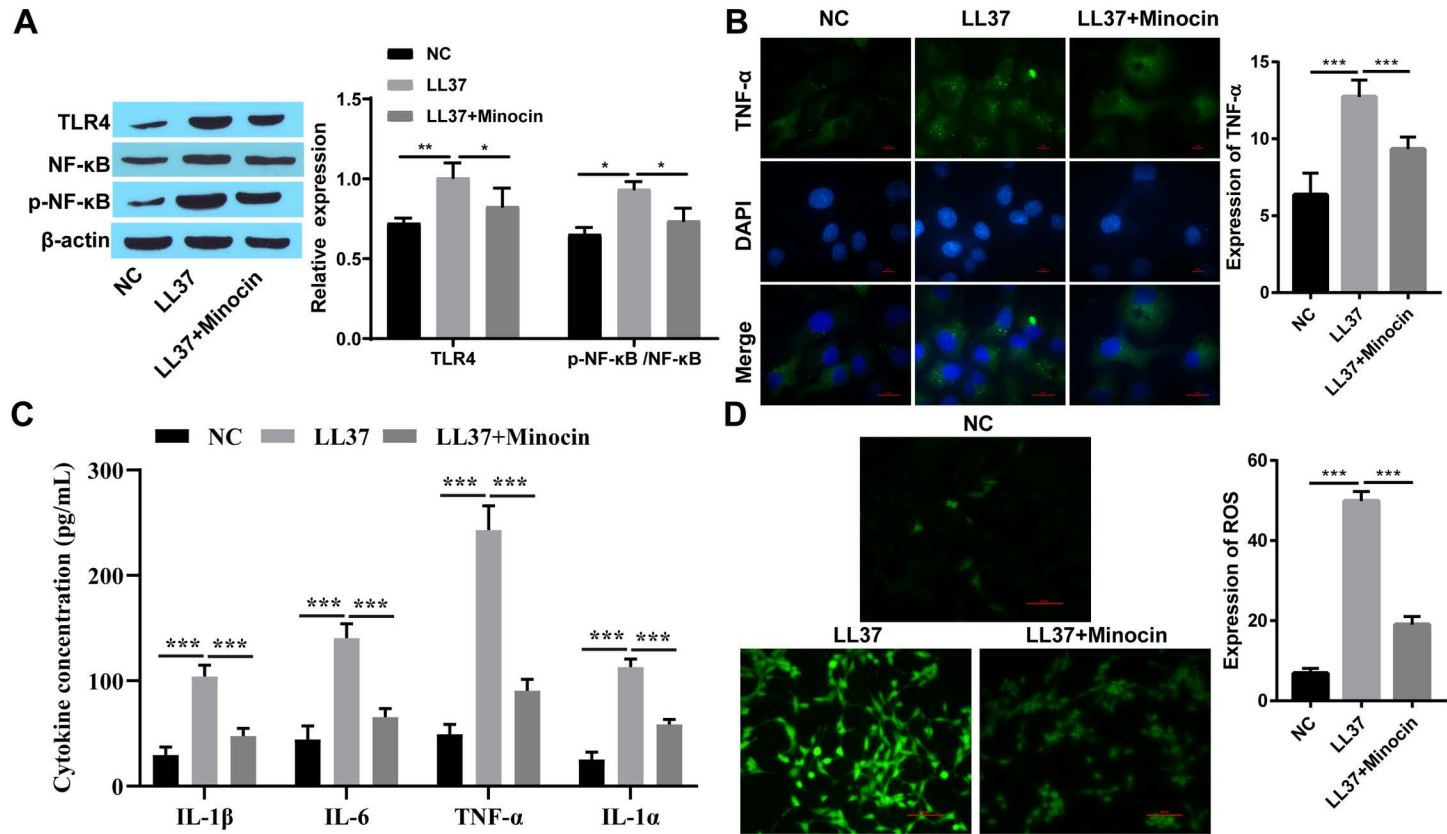

**Fig 1. Minocycline suppressed the expression of NF- κB signaling pathway-associated proteins and downstream inflammatory signaling factors in vitro.** A: The expression of TLR4 and p-NF-κB/NF-κB was detected by western blot; B: The expression of TNF-α was detected by immunofluorescence staining (Scale bar:100μm); C: The levels of the inflammatory cytokines TNF-α, IL-6, IL-1α, and IL-1β were detected by ELISA; D: ROS expression was detected by immunofluorescence (Scale bar:50μm); * P < 0.05, ** P < 0.01, and *** P < 0.001.

were up-regulated after LL37 intervention, which inversely reduced upon minocycline treatment (Figs 1B, 1C 2C, and 2F). ROS production was increased in LL37-treated cells and mice, while decreased after minocycline treatment (Figs 1D and 2D). Moreover, following subcutaneous injection of LL37, marked inflammatory lesions were observed in the dorsal skin of BALB/c mice. Notably, minocycline treatment significantly attenuated these inflammatory responses compared to the LL37-only group (Fig 2A). Similarly, the infiltration of inflammatory cells in BALB/c mice induced by LL37 was increased, which decreased following minocycline intervention (Fig 2E).

### 3.2 Minocycline inhibited rosacea-like inflammation response by regulating TLR4

TLR4 was overexpressed in vivo and in vitro to further explore whether minocycline exerts its anti-inflammatory effect through TLR4. Interestingly, when TLR4 was overexpressed, the efficacy of minocycline was attenuated and the expression of TLR4 was increased. This was accompanied by an increased p-NF-κB/ NF-κB, though the elevation observed in vitro did not reach statistical significance. (Figs 3A and 4B). TNF-α, IL-6, IL-1α, and IL-1β levels were down-regulated following both minocycline and LL37 intervention compared to the LL37 group. However, overexpression of TLR4 weakened the effect of minocycline observing an increase in inflammatory factors including TNF-α, IL-6, IL-1α, and IL -1β (Figs 3B, 3C, 4C, and 4F). Similarly, a significant increase in ROS production was observed while TLR4 was overexpressed

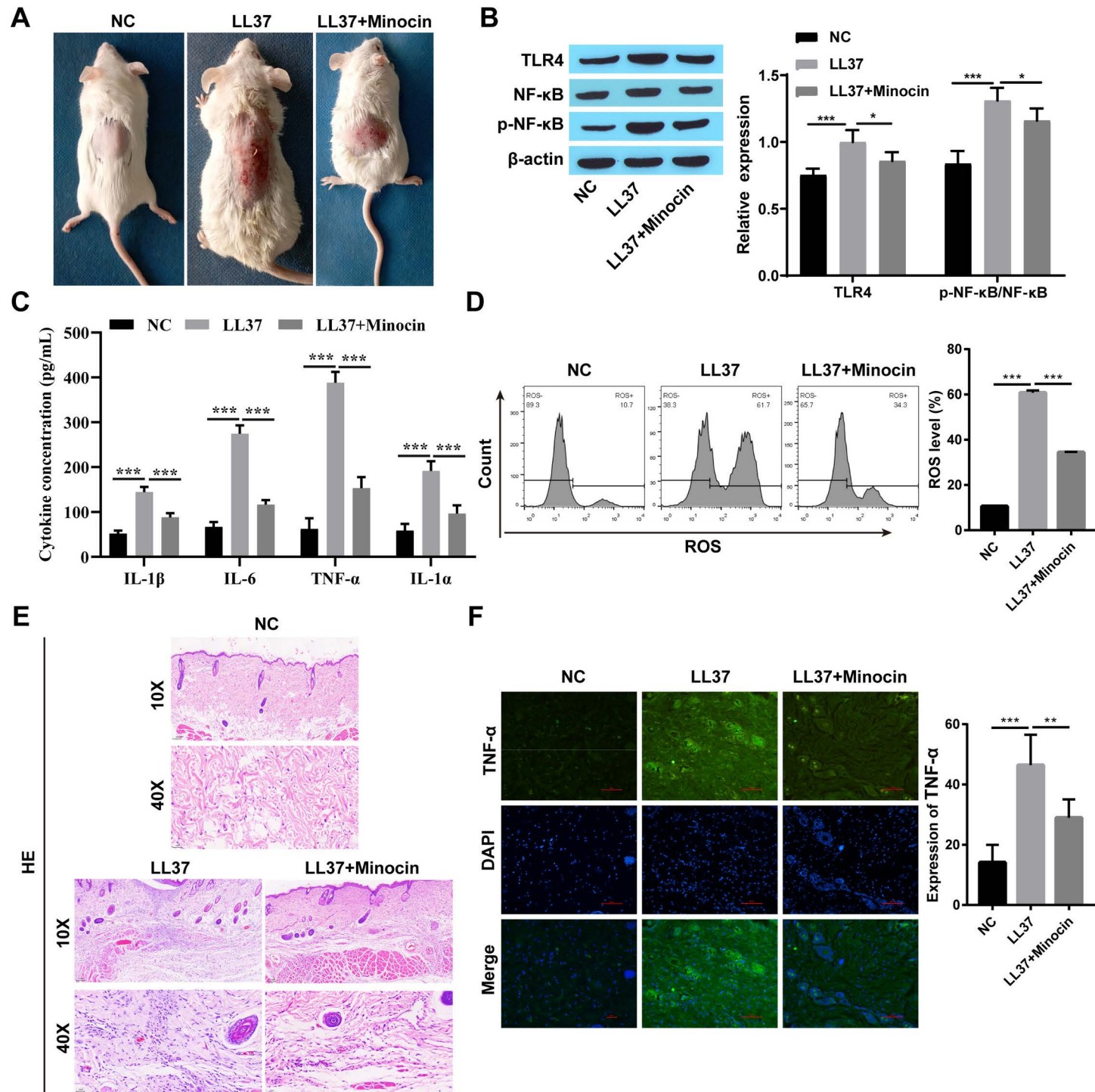

**Fig 2. Minocycline suppressed the expression of NF-κB signaling pathway-associated proteins and downstream inflammatory signaling factors in vivo.** A: The appearance of dorsal skin in three groups of mice; B: The expression of TLR4 and p-NF-κB/NF-κB was detected by western blot; C: The levels of inflammatory factors (TNF-α, IL-6, IL-1α, and IL-1β) were detected by ELISA; D: The level of ROS was detected by flow cytometry; E: Histopathological changes observed with HE staining; F: TNF-α expression was detected by immunofluorescence staining (Scale bar:50μm); * $P < 0.05$, ** $P < 0.01$, and *** $P < 0.001$.

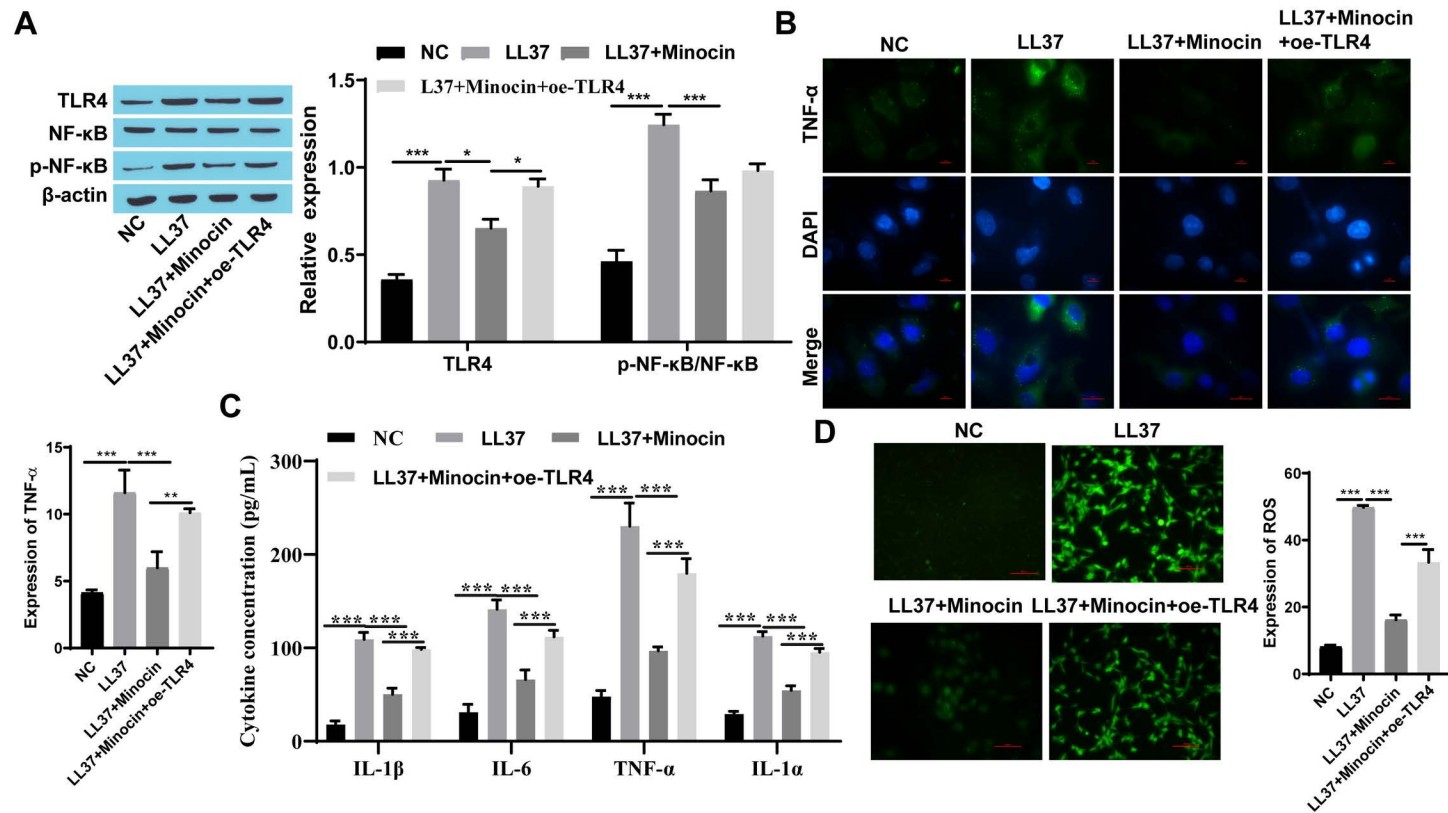

**Fig 3. Minocycline inhibited rosacea-like inflammation response by regulating TLR4 in Hacat cells.** A: The expression of TLR4 and p-NF-κB/NF-κB were detected by western blot; B: The expression of TNF-α was detected by immunofluorescence staining (Scale bar: 100μm); C: Inflammatory cytokines (TNF-α, IL-6, IL-1α, and IL-1β) were detected by ELISA; D: ROS expression was detected by immunofluorescence (Scale bar: 50μm); * P<0.05, *** P<0.001.

(Figs 3D and 4D). When TLR4 was overexpressed, an aggravation of the inflammation manifestation and an increase in inflammatory infiltrating cells were observed in rosacea-like mice treated with minocycline (Figs 4A, and 4E).

### 3.3 Minocycline affects LL37-induced HaCaT cells and rosacea-like mice through the TLR4-mediated NF-κB signaling pathway

Following combined treatment with TLR4 overexpression and the NF-κB signaling pathway inhibitor PDTC, TLR4 expression remained unaltered in rosacea-like cells, while the p-NF-κB/NF-κB was significantly reduced (Fig 5A). In addition, both TLR4 expression and the p-NF-κB/NF-κB showed no marked changes in rosacea-like animal models (Fig 6B). However, the results showed that minocycline treatment decreased levels of inflammatory cytokines like TNF-α, IL-6, IL-1α, and IL-1β. Conversely, TLR4 overexpression increased their levels. Notably, further addition of PDTC attenuated TLR4-overexpression's impact on minocycline and reduced inflammatory cytokine levels (Figs 5B, 5C, 6C, and 6F). Similarly, with the addition of the NF-κB signaling pathway inhibitor PDTC, the impact of TLR4 overexpression on minocycline was attenuated leading to decreased ROS production (Figs 5D, and 6D). Furthermore, the significant alleviation of skin inflammation and the reduced infiltration of inflammatory cells were observed in minocycline-treated rosacea-like mice subjected to TLR4 overexpression and PDTC administration (Figs 6A and 6E).

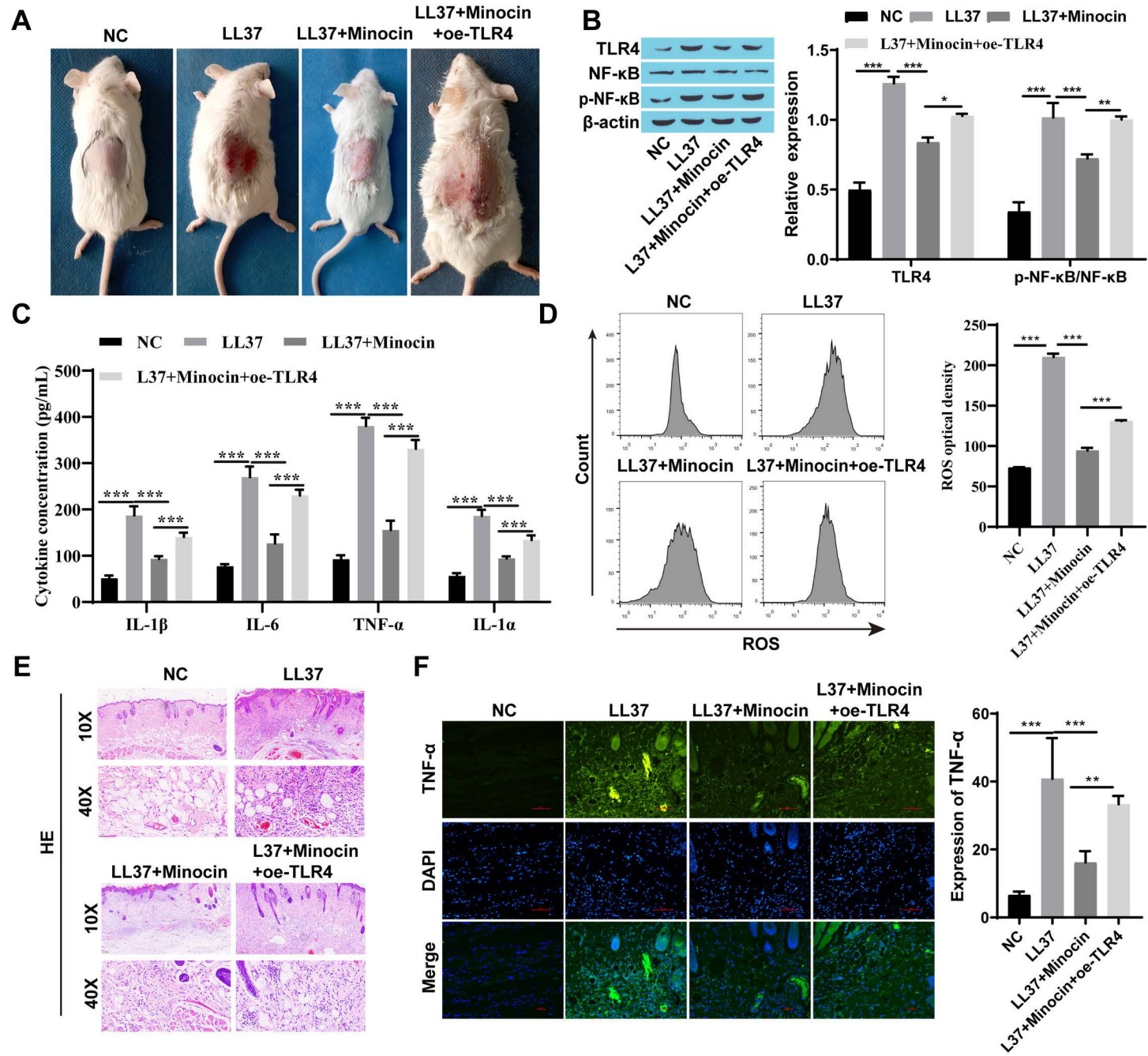

**Fig 4. Minocycline inhibited rosacea-like inflammation response by regulating TLR4 in BALB/c mice.** A: The appearance of dorsal skin in four groups of mice; B: The expression of TLR4 and p-NF-κB/NF-κB was detected by western blot; C: The levels of inflammatory factors (TNF-α, IL-6, IL-1α, and IL-1β) were detected by ELISA; D: The level of ROS was detected by flow cytometry; E: HE-stained skin tissue; F: TNF-α expression was detected by immunofluorescence staining (Scale bar: 100μm); * $P < 0.05$, ** $P < 0.01$, and *** $P < 0.001$.

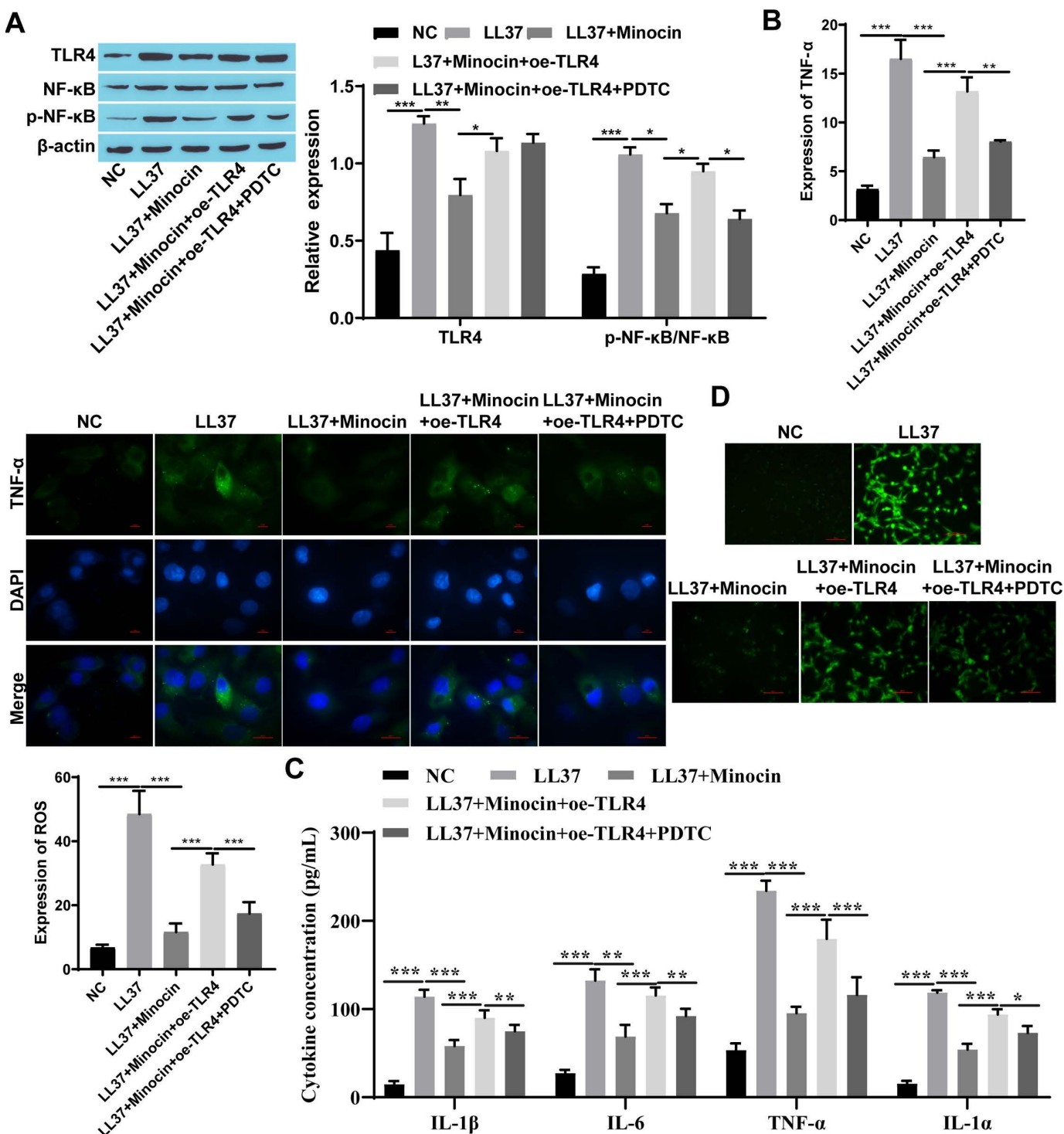

**Fig 5. Minocycline affects LL37-induced HaCaT cells through the TLR4-mediated NF- κB signaling pathway.** A: The expression of TLR4 and NF-κB was detected by western blot; B: The expression of TNF-α was detected by immunofluorescence staining (Scale bar: 100μm); C: The expression of TNF-α, IL-6, IL-1α, and IL-1β was detected by immunofluorescence staining; D: ROS expression was detected by immunofluorescence (Scale bar: 50μm); * $P < 0.05$, ** $P < 0.01$, and *** $P < 0.001$.

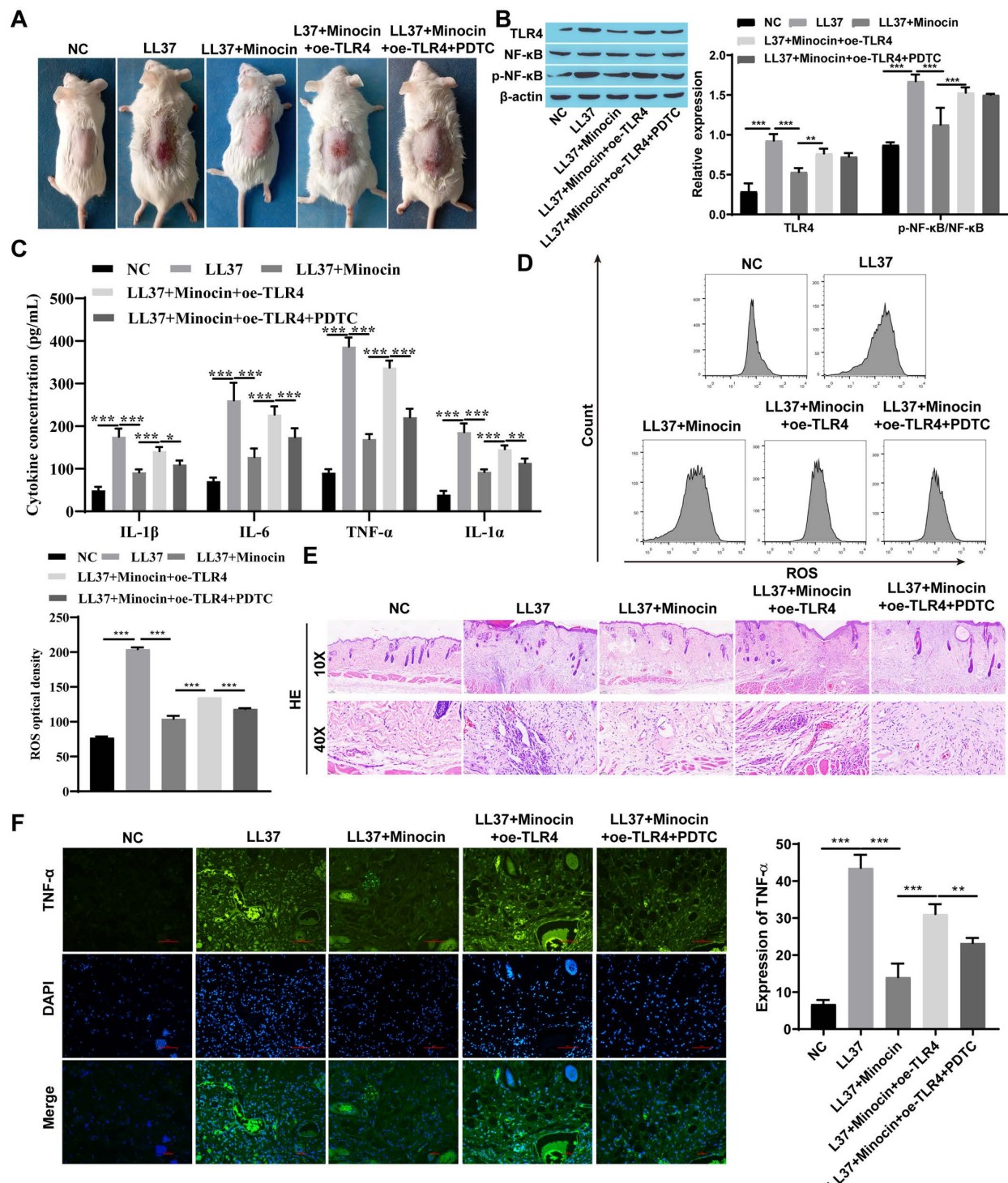

**Fig 6. Minocycline affects rosacea-like mice through the TLR4-mediated NF- κB signaling pathway.** A: The appearance of dorsal skin in four groups of mice; B: The expression of TLR4 and NF-κB was detected by western blot; C: The levels of inflammatory factors (TNF-α, IL-6, IL-1α, and IL-1β) were detected by ELISA; D: ROS were detected by flow cytometry; E: HE staining of skin tissues; F: TNF-α expression was detected by immuno-fluorescence staining (Scale bar: 50μm); * $P<0.05$, ** $P<0.01$, and *** $P<0.001$.

## 4 Discussion

Rosacea is a chronic inflammatory skin disease primarily affecting the midface, exerting a significant impact on patients' psychological well-being and overall quality of life. The pathogenesis of rosacea is complex and varied, in which inflammation plays a pivotal role [4]. Therefore, the administration of anti-inflammatory medications assumes critical importance in its treatment.

Minocycline has been extensively utilized in the treatment of rosacea [19]. However, limited studies exist regarding the specific mechanism of action in this context. The human antibacterial peptide LL37 plays a crucial role in the pathogenesis of rosacea, often used to induce rosacea-like models in vivo and in vitro [20,21].In this study, we explored the possible mechanisms of minocycline treatment of rosacea. The results revealed that minocycline effectively suppressed the expression levels of inflammatory cytokines as well as TLR4 and p-NF-κB in LL-37-treated cells and rosacea-like mice. Notably, overexpression of TLR4 attenuated the inhibitory effect exerted by minocycline. Nevertheless, the addition of the NF-κB signaling pathway inhibitor PDTC based on overexpression of TLR4 significantly restored the efficacy of minocycline.

Toll-like receptors (TLRs) play a crucial role in skin inflammation [22]. Both TLR4 and TLR2 are highly expressed Toll-like receptors in sebaceous glands, regulating genes involved in inflammation, wound healing, and chemotaxis. They play pivotal roles in acne development [23]. Similarly, *Yamasaki K's* study demonstrated an increased expression of TLR2 in rosacea patients, along with elevated levels of tumor necrosis factor-α (TNF-α), the main cytokine induced by TLR signaling [24]. *Wladis EJ et al.* reported increased NF-kB protein levels in eyelid specimens and inflamed skin tissues from patients with rosacea [25]. Reactive oxygen species (ROS), as signaling molecules, participate in diverse biological processes and contribute to cellular homeostasis maintenance [26]. Excessive ROS levels have been reported to mediate inflammatory signaling pathways leading to pathophysiological conditions such as rosacea [27]. TNF-α, IL-6, IL-1α, and IL-1β are well-characterized inflammatory mediators in rosacea pathogenesis, driving key pathological processes including inflammatory cell infiltration, vascular hyperreactivity, and neuroimmune disorder [28,29]. The dysregulation of these cytokines has been consistently associated with disease severity, with elevated expression observed in both lesional skin and serum samples from rosacea patients, making them critical biomarkers for assessing inflammatory progression [29–32]. Consistent with these findings, we demonstrated that TLR4 and p-NF-κB expression, along with inflammatory factor levels, were significantly upregulated, while ROS levels were markedly increased in both rosacea-like cells and mice. Furthermore, Minocycline effectively inhibited the expression of inflammatory cytokines, TLR4, and p-NF-κB in LL-37 treated cells. Minocycline has a significant therapeutic effect on rosacea-like mice, and the inflammatory reaction of skin lesions and inflammatory cell infiltration of mice are significantly improved after intraperitoneal injection of minocycline. The expressions of TLR4 and p-NF-κB in skin tissue were down-regulated, and the levels of TNF-α, IL-6, IL-1α, IL-1β, and ROS were decreased.

Previous research has shown that NF-κB signaling is associated with various inflammatory skin diseases and observed significant activation during dermatitis exacerbation [33]. TLR4 can regulate infection induction or the inflammatory response through endogenous molecules and apoptotic processes [34,35]. The TLR4-NF-κB pathway is the main regulatory pathway involved in cellular inflammation [36]. Additionally, a study revealed that Dendrobium polysaccharides could inhibit the activation of the TLR4/NF-κB pathway in mouse skin, thereby reducing inflammation and exerting a protective effect against rosacea in mice [37]. In our study, Minocycline significantly downregulated the expression of TLR4 and inflammatory cytokines while inhibiting ROS production. However, when TLR4 was overexpressed, minocycline's effect was weakened. The efficacy of minocycline was substantially restored upon the addition of the NF-κB signaling pathway inhibitor PDTC. Therefore, we propose that minocycline suppresses NF-κB pathway activation by downregulating TLR4, thereby reducing the expression of inflammatory cytokines and inhibiting ROS production.

Our findings indicate that minocycline modulates rosacea development via the TLR4-mediated NF-κB signaling pathway. Overexpression of TLR4 stimulates cellular inflammation, thwarting the therapeutic effect of minocycline. The TLR4 receptor is capable of regulating downstream NF-κB signaling activity. NF-κB phosphorylation regulates the transcriptional activity of NF-κB in the nucleus, triggering the production of inflammatory cytokines [38]. In this study, HaCaT cells were treated with the NF-κB signaling pathway inhibitor PDTC and found that inhibition of NF-κB activity attenuated the impact of TLR4 overexpression while suppressing inflammation and ROS levels. These results suggest that minocycline suppresses inflammation, reduces ROS production, and alleviates inflammatory progression in rosacea by downregulating TLR4 expression and inhibiting the NF-κB signaling pathway.

However, since we did not directly validate the relationship between TLR4 and NF-κB in HaCaT cells, our study provides evidence for an association rather than causation. Future investigations should be conducted to elucidate specific mechanisms underlying minocycline-induced downregulation of TLR4 expression in rosacea.

Furthermore, It is critical to emphasize that minocycline's therapeutic effects on rosacea likely extend beyond TLR4/NF-κB modulation. Preclinical evidence for other diseases indicates its capacity to inhibit matrix metalloproteinases (MMPs) [39–41], regulate macrophage polarization [42], and exert antioxidant/neuroprotective effects [43–45]. While these pleiotropic mechanisms could synergistically contribute to the observed anti-inflammatory outcomes, the precise mechanistic contributions require further investigation.

## 5 Conclusion

Minocycline alleviates inflammation progression in rosacea by downregulating TLR4 and inhibiting the activation of the NF-κB pathway, providing a scientific basis for subsequent clinical treatment.

## Author contributions

**Conceptualization:** Hua Gu.

**Data curation:** Ying Tu, Hua Gu.

**Formal analysis:** Piyan Hua, Zhenghui Yang, Yunting He, Qiuyan Yao.

**Funding acquisition:** Hua Gu.

**Investigation:** Piyan Hua, Zhenghui Yang, Yunting He, Qiuyan Yao.

**Methodology:** Piyan Hua, Zhenghui Yang, Yunting He, Qiuyan Yao.

**Supervision:** Ying Tu, Li He.

**Validation:** Zhenghui Yang, Yunting He.

**Visualization:** Piyan Hua, Ying Tu.

**Writing – original draft:** Piyan Hua.

**Writing – review & editing:** Piyan Hua, Hua Gu.

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
