## [Decision Letter · Decision Letter 0]

16 Jan 2025

PONE-D-24-43618Minocycline inhibits rosacea-like inflammation through the TLR4-mediated NF-κB signaling pathway in vivo and vitroPLOS ONE

Dear Dr. Gu,

Thank you for submitting your manuscript to PLOS ONE. After careful consideration, we feel that it has merit but does not fully meet PLOS ONE’s publication criteria as it currently stands. Therefore, we invite you to submit a revised version of the manuscript that addresses the points raised during the review process.

The manuscript has been evaluated by three reviewers, and their comments are available below.

The reviewers have raised a number of concerns that need attention. In particular, they request additional information on methodological aspects of the study, the inclusion of images to show the changes in the macroscopic appearance of skin lesions, and further discussion.

Could you please revise the manuscript to carefully address the concerns raised?

We look forward to receiving your revised manuscript.

Kind regards,

Helen Howard

Staff Editor

PLOS ONE

Journal Requirements:

3. Thank you for stating the following financial disclosure: This work was supported by the Yunnan Revitalization Talent Support Program.Hua Gu is the Fund recipient.  

4. In the online submission form, you indicated that the data that support the findings of this study are available from the corresponding author upon reasonable request. 

Reviewers' comments:

Reviewer's Responses to Questions

**Comments to the Author**

1. Is the manuscript technically sound, and do the data support the conclusions?

Reviewer #1: Yes

Reviewer #2: Yes

Reviewer #3: Yes

2. Has the statistical analysis been performed appropriately and rigorously? 

Reviewer #1: Yes

Reviewer #2: N/A

Reviewer #3: Yes

3. Have the authors made all data underlying the findings in their manuscript fully available?

Reviewer #1: Yes

Reviewer #2: Yes

Reviewer #3: Yes

4. Is the manuscript presented in an intelligible fashion and written in standard English?

Reviewer #1: Yes

Reviewer #2: Yes

Reviewer #3: Yes

5. Review Comments to the Author

Reviewer #1: The authors conducted in-vitro and in-vivo experiments to demonstrate the effect of minocycline on the TLR4/NF-kB pathway in LL37-induced models of rosacea. The results showed that minocycline downregulated TLR4 and inhibited NF-kB pathway activity, proposing a molecular basis for the efficacy of minocycline in treating rosacea. This study adds to our understanding of the mechanism of action of minocycline in rosacea. Thank you for including me in the peer review of this manuscript, please see my comments below:

1. Introduction: “However, the 34 etiology and pathogenesis of rosacea remain have not been fully elucidated”. Remove “remain”.

2. Introduction: “immunoregulatory, neuroprotective and angiogenic properties”. Angiogenic or anti-angiogenic? Given the clinical features of rosacea, it would seem that anti-angiogenic properties would be beneficial.

3. Introduction: “TLR4/p38 MAPK and NF κB pathways”. Abbreviations not defined. Several other abbreviations also not defined at first appearance, e.g. IKK, NC, IL, TNF, etc.

4. Discussion: “Toll-like receptors (TLRs) playing a crucial role in skin inflammation”. Change “playing” to “play”.

5. Discussion: “YAMASAKI K’ study”. This does not need to be in all capitals.

6. Discussion: “Future investigations should be conducted to elucidate specific mechanisms 310 underlying minocycline-induced upregulation of TLR4 expression in rosacea.” Upregulation or downregulation?

7. Why were TNF-α, IL-6, IL-1α, and IL-1β chosen to be measured? If these cytokines have been shown to be particularly important in rosacea pathogenesis, please introduce and cite references in the introduction to explain the rationale.

8. The term “minocin” is used instead of “minocycline” throughout the results and other places in the manuscript. Minocin is a brand name.

9. While the results demonstrate the effects of minocycline on the TLR4/NK-kB pathway, minocycline is not a specific inhibitor of this pathway. Another limitation could be potential confounding from inhibition of other pro-inflammatory factors by minocycline. Please briefly discuss other potential targets of minocycline.

Reviewer #2: This is a good and orginal study calling attention to the pathogenesis and treatment mechanisms of acne rosacea. The manuscript is written well and on a scientific base. However, there are some grammatical mistakes in the main text that need to be corrected. Thanks to the authors for their work.

Reviewer #3: 1. "Minocycline-only group" was not included in the control settings of all experiments.

2. In the in vivo experimental results, no images were provided to show the changes in the macroscopic appearance of skin lesions. Based on the current results, it is difficult to confirm whether the model was successfully established and whether the minocycline treatment was effective.

6. PLOS authors have the option to publish the peer review history of their article (what does this mean? ). If published, this will include your full peer review and any attached files.

**Do you want your identity to be public for this peer review?** For information about this choice, including consent withdrawal, please see our Privacy Policy .

Reviewer #1: No

Reviewer #2: No

Reviewer #3: No

---

## [Author Response · Author response to Decision Letter 1]

25 Mar 2025

Responses to Journal Requirements

Comment 1: Please ensure that your manuscript meets PLOS ONE's style requirements, including those for file naming.

Response: Thank you for highlighting this important requirement. We have carefully reviewed and adjusted the manuscript to comply with PLOS ONE's style guidelines.

Comment 2: To comply with PLOS ONE submissions requirements, in your Methods section, please provide additional information regarding the experiments involving animals and ensure you have included details on (1) methods of sacrifice, (2) methods of anesthesia and/or analgesia, and (3) efforts to alleviate suffering.

Response: In order to comply with the requirements submitted by PLOS ONE, the completed modeling methods are provided in the Methods section, supplemented with additional information about animal experiments, including sacrificial methods, anesthesia and/or analgesia methods, and pain relief efforts. The supplementary content has been marked in red font in the article.

‘Seven-week-old BALB/c mice were purchased from Shanghai SLAC Experimental Animal Co. Ltd. (Shanghai). All animal experiments were carried out under specific pathogen-free conditions. In the process of feeding mice, keep the feeding environment clean, quiet, temperature (22 ± 2) ℃, humidity (50 ± 5) % constant, provide adequate food and clean drinking water, to reduce the stress and pain caused by external environmental factors on mice. All studies and experimental procedures were approved by the Animal Ethics Committee of The First Affiliated Hospital of Kunming Medical University. The mice were divided into five groups (the NC group, the LL37 group, the LL37+Minocin group, the LL37+Minocin+oe-TLR4 group, and the LL37+Minocin+oe-TLR4+PDTC group) and treated differently. The mice were shaved 24 hours before treatment and then injected subcutaneously with 40 μL of LL37 peptide twice a day for 3 days. The mice in the LL37+Minocin group were administered an intraperitoneal injection of minocycline at a dose of 50 mg/kg/day for 3 days. For the LL37+Minocin+oe-TLR4 group, Minocin-treated mice were injected with 100 μg of oe-TLR4 plasmid via the tail vein. For the LL37+Minocin+oe-TLR4+PDTC (5108–96-3, AbMole) group, the mice were intraperitoneally injected with PDTC at a dose of 50 mg/kg. The experiment was performed 72 hours after the initial injection of LL37. In order to alleviate the pain of mice, mice were anesthetized by intraperitoneal injection of 1% sodium pentobarbital (40mg/kg) before the experimental operation. After the experiment, the mice were euthanized by cervical dislocation. Skin inflammation was evaluated by the severity of erythema and edema. Skin tissues were collected for protein extraction, hematoxylin-eosin staining, and immunofluorescence analysis.’

Comment 3: Thank you for stating the following financial disclosure: This work was supported by the Yunnan Revitalization Talent Support Program. Hua Gu is the Fund recipient.

Response: Thank you for raising this important point. We have revised the manuscript to explicitly clarify the funders’ role in the Funding section: The funders had no role in study design, data collection, and analysis, decision to publish, or preparation of the manuscript.

Comment 4: In the online submission form, you indicated that the data that support the findings of this study are available from the corresponding author upon reasonable request. All PLOS journals now require all data underlying the findings described in their manuscript to be freely available to other researchers, either 1. In a public repository, 2. Within the manuscript itself, or 3. Uploaded as supplementary information. This policy applies to all data except where public deposition would breach compliance with the protocol approved by your research ethics board. If your data cannot be made publicly available for ethical or legal reasons (e.g., public availability would compromise patient privacy), please explain your reasons on resubmission and your exemption request will be escalated for approval.

Response: Thank you for your comments. We have complied with the journal’s requirements by uploading all relevant data associated with this study to the Figshare public repository and have explicitly stated the data access methods in the revised manuscript. The specific updates are detailed below:

Data Storage and Access

All raw data have been uploaded to Figshare (DOI: 10.6084/m9.figshare.28464782). The dataset has been set to public access, ensuring that readers can freely download and use the materials.

Manuscript Updates

In the "Data Availability Statement" section of the manuscript, we have added the following declaration:

“All relevant data from this study has been uploaded to Figshare and can be accessed at https://figshare.com/s/659c253d0b649844765f. This dataset is licensed under CC BY 4.0, allowing free reuse with attribution. (DOI: 10.6084/m9.figshare.28464782)”

Comment 5: Your ethics statement should only appear in the Methods section of your manuscript. If your ethics statement is written in any section besides the Methods, please move it to the Methods section and delete it from any other section. Please ensure that your ethics statement is included in your manuscript, as the ethics statement entered into the online submission form will not be published alongside your manuscript.

Response: Thank you for your valuable comments. We have carefully revised the manuscript according to your suggestions. The ethics statement has been moved exclusively to the Methods section (specifically in the Animal). All duplicate ethics statements previously appearing in DECLARATION have been completely removed.

Response to reviewer1

Comment 1-6: 1. Introduction: “However, the 34 etiology and pathogenesis of rosacea remain have not been fully elucidated”. Remove “remain”. 2. Introduction: “immunoregulatory, neuroprotective and angiogenic properties”. Angiogenic or anti-angiogenic? Given the clinical features of rosacea, it would seem that anti-angiogenic properties would be beneficial. 3. Introduction: “TLR4/p38 MAPK and NF κB pathways”. Abbreviations not defined. Several other abbreviations also not defined at first appearance, e.g. IKK, NC, IL, TNF, etc. 4. Discussion: “Toll-like receptors (TLRs) playing a crucial role in skin inflammation”. Change “playing” to “play”. 5. Discussion: “YAMASAKI K’ study”. This does not need to be in all capitals. 6. Discussion: “Future investigations should be conducted to elucidate specific mechanisms 310 underlying minocycline-induced upregulation of TLR4 expression in rosacea.” Upregulation or downregulation?

Response: We would like to express our sincere gratitude for your positive acknowledgment of our work and constructive comments. Regarding the grammatical and typographical issues identified in the Introduction and Discussion sections, we fully concur with your insightful recommendations and have accordingly implemented the suggested amendments throughout the manuscript.

Comment 7: Why were TNF-α, IL-6, IL-1α, and IL-1β chosen to be measured? If these cytokines have been shown to be particularly important in rosacea pathogenesis, please introduce and cite references in the introduction to explain the rationale.

Response: We sincerely appreciate the reviewer’s insightful comment. The selection of TNF-α, IL-6, IL-1α, and IL-1β as key biomarkers in this study was based on their well-established roles in rosacea pathogenesis, particularly in driving inflammation, vascular dysfunction, and immune dysregulation. We have added the corresponding introduction and citations in the Discussion sections, see the red font section of the manuscript.

TNF-α, IL-6, IL-1α, and IL-1β are well-characterized inflammatory mediators in rosacea pathogenesis, driving key pathological processes including inflammatory cell infiltration, vascular hyperreactivity, and neuroimmune disorder [28, 29]. The dysregulation of these cytokines has been consistently associated with disease severity, with elevated expression observed in both lesional skin and serum samples from rosacea patients, making them critical biomarkers for assessing inflammatory progression [29-32].

Comment 8: The term “minocin” is used instead of “minocycline” throughout the results and other places in the manuscript. Minocin is a brand name.

Response: We sincerely appreciate the reviewer's meticulous attention to terminology consistency. In accordance with scientific writing standards requiring the use of generic drug names, we have systematically replaced "Minocin" with "minocycline" throughout the Results, Discussion, and other sections of the manuscript. The initial use of "Minocin" in the Methods section was intentionally retained to specify the exact commercial formulation used in our experiments, as drug bioavailability and excipient composition may vary between brands. All instances of "Minocin" outside the Methods section have now been revised to "minocycline" to maintain scientific objectivity. We have additionally performed a full-text verification to ensure no unintended usage remains.

Comment 9: While the results demonstrate the effects of minocycline on the TLR4/NK-kB pathway, minocycline is not a specific inhibitor of this pathway. Another limitation could be potential confounding from inhibition of other pro-inflammatory factors by minocycline. Please briefly discuss other potential targets of minocycline.

Response: We sincerely thank the reviewer for raising this critical point regarding the multifunctional nature of minocycline. As rightly noted, while our study focused on its effects on the TLR4/NF-κB pathway, we fully acknowledge that minocycline is not a pathway-specific inhibitor and may exert broader anti-inflammatory actions. In the revised Discussion section, we have added the following content to address this limitation:

" Furthermore, It is critical to emphasize that minocycline's therapeutic effects on rosacea likely extend beyond TLR4/NF-κB modulation. Preclinical evidence for other diseases indicates its capacity to inhibit matrix metalloproteinases (MMPs)[39-41], regulate macrophage polarization[42], and exert antioxidant/neuroprotective effects[43-45]. While these pleiotropic mechanisms could synergistically contribute to the observed anti-inflammatory outcomes, the precise mechanistic contributions require further investigation."

Response to Reviewer 2

Comment: This is a good and original study calling attention to the pathogenesis and treatment mechanisms of acne rosacea. The manuscript is written well and on a scientific base. However, there are some grammatical mistakes in the main text that need to be corrected. Thanks to the authors for their work.

Response: We sincerely appreciate the reviewer's encouraging evaluation of our work and their keen attention to manuscript quality. We have conducted a full-scale grammatical revision of the manuscript with Grammarly Premium.

Response to Reviewer 3

Comment 1: "Minocycline-only group" was not included in the control settings of all experiments.

Response: We sincerely appreciate your meticulous review of our manuscript and the valuable comments you've provided. Regarding your concern about the absence of a "Minocycline-only group" in the control settings of all experiments, we would like to present the following considerations.

Firstly, the core objective of this study is to explore the effect of Minocycline or its combination with other intervention factors (e.g.oe-TLR4, PDTC) on disease progression in the LL37-induced rosacea-like cell and animal model, and to clarify the mechanism of action of TLR4 and NF-κB signaling pathways. However, the minocycline-alone group was not included in the experimental design because it was not associated with the core purpose of the study.

Secondly, a number of targeted control groups have been set up in the current experiment, such as the NC group as the normal physiological state control, the LL37 group as the disease model control, and the LL37 + Minocycline group to evaluate the role of Minocycline in the disease model. These control groups can fully meet the logical needs of experimental design and provide a reliable reference for the research results.

Finally, considering the resource and time constraints in actual scientific research, the additional Minocycline single-use group will significantly increase the experimental cost and time investment. On the basis of ensuring that the research objectives can be achieved, the experimental design has been optimized to achieve the efficient use of resources. The existing control group was set up to meet the needs of the study, so the Minocycline-only group was not included.

Comment 2: In the in vivo experimental results, no images were provided to show the changes in the macroscopic appearance of skin lesions. Based on the current results, it is difficult to confirm whether the model was successfully established and whether the minocycline treatment was effective.

Response: We sincerely appreciate your insightful review and constructive feedback on our manuscript. We fully understand your concerns regarding the lack of images depicting the macroscopic appearance changes of skin lesions in the in-vivo experimental results. To address this issue, we have now added a set of high-quality images in the revised manuscript. These images clearly show the skin lesions in different groups. From these images, it is evident that in the LL37 group, there are obvious signs of erythema, papule, and other typical manifestations of the established model, which strongly indicates the successful establishment of the rosacea-like model. In the LL37 + Minocycin group, a significant improvement in the macroscopic appearance of the skin lesions can be observed compared to the LL37 group, demonstrating the effectiveness of Minocycline treatment. Please see Figures 2A, 4A, 6A. We believe that these added images will greatly enhance the comprehensibility and persuasiveness of our study. Once again, we are grateful for your valuable suggestions, which have helped us to improve the quality of our manuscript.

---

## [Decision Letter · Decision Letter 1]

11 Apr 2025

Minocycline inhibits rosacea-like inflammation through the TLR4-mediated NF-κB signaling pathway in vivo and in vitro

PONE-D-24-43618R1

Dear Dr. Gu,

We’re pleased to inform you that your manuscript has been judged scientifically suitable for publication and will be formally accepted for publication once it meets all outstanding technical requirements.

Kind regards,

Divakar Sharma

Academic Editor

PLOS ONE

Additional Editor Comments (optional):

Accept

Reviewers' comments:

Reviewer's Responses to Questions

**Comments to the Author**

1. If the authors have adequately addressed your comments raised in a previous round of review and you feel that this manuscript is now acceptable for publication, you may indicate that here to bypass the “Comments to the Author” section, enter your conflict of interest statement in the “Confidential to Editor” section, and submit your "Accept" recommendation.

Reviewer #1: All comments have been addressed

Reviewer #4: All comments have been addressed

2. Is the manuscript technically sound, and do the data support the conclusions?

Reviewer #1: Yes

Reviewer #4: Yes

3. Has the statistical analysis been performed appropriately and rigorously? 

Reviewer #1: Yes

Reviewer #4: Yes

4. Have the authors made all data underlying the findings in their manuscript fully available?

Reviewer #1: Yes

Reviewer #4: Yes

5. Is the manuscript presented in an intelligible fashion and written in standard English?

Reviewer #1: Yes

Reviewer #4: Yes

6. Review Comments to the Author

Reviewer #1: I would like to thank the authors for their efforts in further elucidating the pathogenesis mechanisms of rosacea. My comments have been addressed satisfactorily.

Reviewer #4: This is a revised version of manuscript reviewed by other experts, I have went through the comments and the reponses and I beleive authors have made satisfactory revison. Now this manuscript can be accepted in PLOS One.

7. PLOS authors have the option to publish the peer review history of their article (what does this mean? ). If published, this will include your full peer review and any attached files.

**Do you want your identity to be public for this peer review?** For information about this choice, including consent withdrawal, please see our Privacy Policy .

Reviewer #1: No

Reviewer #4: No

---

## [Editor Report · Acceptance letter]

PONE-D-24-43618R1

PLOS ONE

Dear Dr. Gu,

I'm pleased to inform you that your manuscript has been deemed suitable for publication in PLOS ONE. Congratulations! Your manuscript is now being handed over to our production team.

Kind regards,

on behalf of

Dr. Divakar Sharma

Academic Editor

PLOS ONE